# Targeting Hydroxybenzoic Acids to Mitochondria as a Strategy to Delay Skin Ageing: An In Vitro Approach

**DOI:** 10.3390/molecules27196183

**Published:** 2022-09-21

**Authors:** Carlos Fernandes, Fernando Cagide, Jorge Simões, Carlos Pita, Eurico Pereira, Afonso J. C. Videira, Pedro Soares, José F. S. Duarte, António M. S. Santos, Paulo J. Oliveira, Fernanda Borges, Filomena S. G. Silva

**Affiliations:** 1Mitotag, Biocant Park, Parque Tecnológico de Cantanhede, Núcleo 04, Lote 4, 3060-197 Cantanhede, Portugal; 2CIQUP-IMS/Department of Chemistry and Biochemistry, Faculty of Sciences, University of Porto, 4169-007 Porto, Portugal; 3CNC-Center for Neuroscience and Cell Biology, CIBB-Centre for Innovative Biomedicine and Biotechnology, University of Coimbra, 3004-504 Coimbra, Portugal

**Keywords:** hydroxybenzoic acid-based mitochondria-targeted antioxidants, skin ageing, senescence, oxidative stress, mitochondria

## Abstract

Targeting antioxidants to mitochondria is considered a promising strategy to prevent cellular senescence and skin ageing. In this study, we investigate whether four hydroxybenzoic acid-based mitochondria-targeted antioxidants (MitoBENs, MB1-4) could be used as potential active ingredients to prevent senescence in skin cells. Firstly, we evaluated the chemical stability, cytotoxicity, genotoxicity and mitochondrial toxicity of all compounds. We followed this by testing the antioxidant protective capacity of the two less toxic compounds on human skin fibroblasts. We then assessed the effects of the best hit on senescence, inflammation and mitochondrial remodeling on a 3D skin cell model, while also testing its mutagenic potential. Cytotoxicity and mitochondrial toxicity rankings were produced: MB3 < MB4 ≃ MB1 < MB2 and MB3 < MB1 < MB4 < MB2, respectively. These results suggest that pyrogallol-based compounds (MB2 and MB4) have lower cytotoxicity. The pyrogallol derivative, MB2, containing a 6-carbon spacer, showed a more potent antioxidant protective activity against hydrogen peroxide cytotoxicity. In a 3D skin cell model, MB2 also decreased transcripts related to senescence. In sum, MB2’s biological safety profile, good chemical stability and lack of mutagenicity, combined with its anti-senescence effect, converts MB2 into a good candidate for further development as an active ingredient for skin anti-ageing products.

## 1. Introduction

Skin ageing is a complex biological process affected by genetic and environmental factors, leading to a progressive loss of structure and function [1,2]. During skin ageing, the skin undergoes degenerative changes, including loss of regenerative capacity and elasticity, and epidermal/dermal thinning that results in skin wrinkling and dryness, as well as experiencing an accumulation of senescent cells [3,4,5].

The aged skin is characterized by decreased cell replacement, the decline of mitochondrial function, accumulation of mitochondrial DNA (mtDNA) deletions, and an overproduction of reactive oxygen species (ROS), which, among other events, are observed in both the dermal and epidermal layers. Generally, the proximity between the electron transport chain and mtDNA makes the mitochondrial genome extremely susceptible to mutations, disrupting the oxidative phosphorylation (OXPHOS) system and compromising mitochondrial function [3]. In a continuous cycle, mitochondrial dysfunction leads to increased mitochondrial ROS production with concomitant oxidative damage to mitochondria [6]. Although the skin epidermis possesses an efficient endogenous antioxidant defence network, including different types of antioxidant enzymes, such as superoxide dismutase, and catalase, as well as other types of antioxidants, including α-tocopherol, ascorbate, and glutathione [7,8] the excessive production of ROS can cause a redox imbalance that results in oxidative damage of biomolecules, such as proteins, lipids, and DNA, in skin cell [9] processes that can, consequently, cause cell senescence or death. 

Potential interventions to either prevent or inhibit senescence in skin models have been reported [10,11]. Different senescence markers have also been used to measure outcomes of anti-senescence interventions, including the expression of matrix metalloproteinases (MMPs) that alter the extracellular matrix and play a significant role in wrinkle formation [12], or the expression of TIMP metallopeptidase inhibitor 1 (TIMP-1), which inhibits the most relevant MMPs involved in the degradation of various proteins in the extracellular matrix, reducing the progression of senescence [13]. Antioxidants are a promising strategy to decrease oxidative stress (OS) normally associated with skin senescence and death, [14,15]. Several types of exogenous antioxidants of different chemical classes (e.g., Vitamin E, Vitamin C, Coenzyme Q10, Phenolic acids), as well as their derivatives (e.g: Tocopheryl acetate, γ-Tocotrienol, Ascorbyl-6-palmitate), can be incorporated in anti-skin ageing products to reduce the harmful effects of oxidative damage and to delay skin ageing [16,17,18,19]. However, as these antioxidants do not directly target mitochondria, their effects in this organelle are reduced [20,21]. This is a drawback related to their inherent low bioavailability, which limits their capacity to reach the mitochondrial matrix [22,23].

Mitochondria-targeted antioxidants have been designed by several groups using different antioxidant cores, including the coenzyme Q and plastoquinone (MitoQ and SKQ, respectively) [24,25] and hydroxycinnamic [26,27] and hydroxybenzoic acids [28]. In general, in this process, the antioxidant core is linked to a carrier (triphenylphosphonium (TPP) lipophilic cation) through an alkyl linker. These types of mitochondriotropic antioxidants allow the mitochondrial uptake of the antioxidant core, due to the negatively-charged mitochondrial transmembrane electric potential (∆ψ) [29,30,31]. Despite the promising results of MitoQ and SkQ1 in preventing mitochondrial oxidative damage in different cellular models (HeLa cells, Jurkat cells, isolated liver mitochondria, and others) [29,32], these two antioxidants have shown significant cytotoxic effects in different in vitro neuronal and hepatic cell-based models [33]. They can act as prooxidants disrupting the mitochondrial membrane integrity at concentrations above 1 μM [25,34].

Using a similar rational design, hydroxybenzoic acids were used as the antioxidant core and coupled to a TPP cation through an alkyl chain with a variable lenght [28,35]. This resulted in a library of novel molecules named MitoBENs (MBs, Figure 1A) based on protocatechuic (MB1 and MB3, Figure 1B) and gallic acid (MB2 and MB4, Figure 1B) [28]. The studies performed so far revealed that MB1 and MB2 showed low cytotoxic effects on rat cardiomyoblasts (H9c2), human neonatal dermal fibroblasts (NHDF), and human hepatoma (HepG2) cells [28], as well as cytoprotective effects against tert-butyl hydroperoxide (t-BHP) insults in both H9c2 and NHDF cells [28]. Teixeira et al. [36] also showed that MB2 decreased hydrogen peroxide (H_2_O_2_)-induced cell death in primary human skin fibroblasts, stimulating the expression of the nuclear factor erythroid 2-Related Factor 2 (NRF2; a master regulator of the cellular antioxidant response toward OS), activating endogenous ROS-protective pathways.

Despite these results, the potential of the new mitochondria-targeted agents to prevent fibroblasts’ senescence is still undetermined. To uncover these biological effects, we examined the effect of MitoBENs on cell senescence-related transcripts in a 3D skin model for the first time. Firstly, we evaluated the chemical stability of the four MitoBENs tested here (MB1, MB2, MB3, and MB4, Figure 1A), followed by investigating their toxicity on normal human dermal fibroblasts (NHDF) cells (cell viability, ATP content, mitochondrial toxicity and genotoxicity). A comparison was made with parental compounds, and with MitoQ and SkQ1, which are well-characterized mitochondria-targeted antioxidants, now having commercial applications in cosmetic products [37,38]. After this preliminary screening, MitoBENs that showed low cytotoxicity were investigated for their protective effects against H_2_O_2_ cytotoxicity on 2D cell cultures. MB2 was singled out as the best molecule from the initial set of experiments, in which it was tested on a 3D skin model for its toxicity and effects on metabolic, inflammation or senescence markers. The results of our work provide relevant data to validate MB2 as an active ingredient to be used in topical skin products.

## 2. Results

### 2.1. Chemistry 

The mitochondriotropic antioxidants (MB1–MB4) (Figure 1) and their respective intermediates were synthesized according to the strategy described by Oliveira et al. [28,39]. The chemical modifications in MitoBENs involved the variation of the number of phenolic groups on the aromatic ring (catecholic: MB1 and MB3 or pyrogallol: MB2 or MB4) and the length of the alkyl linker between the carboxamide and TPP cation (6-carbon spacer: MB1 and MB2 vs. 8-carbon spacer: MB3 and MB4), as shown in Figure 1. 

The structural characterization was performed by NMR (1H, 13C and DEPT) and mass spectroscopy (EI-MS and ESI-MS) techniques, thereby confirming their chemical identity. The target compounds were obtained in moderate yields (>49%) with a purity grade higher than 97%, determined through UHPLC, using the experimental conditions described in Section 4.2.3.

### 2.2. Stability of MitoBENs 

The stability of the mitochondriotropic antioxidants (MB1–MB4) was determined by exposing aqueous solutions (pH 7.4) to indoor light conditions for 7 days. Aliquots were collected and analyzed at predetermined time-points by reverse-phase ultra-high-performance liquid chromatography (RP-UHPLC). This method has been widely used due to its low cost and high sensitivity [40]. The percentage of MB compounds (%PRC) at each time-point is shown in Figure 2. In general, MitoBENs showed good stability (%PRC > 85%) after 7 days of light exposure, except for MB4, which showed a decrease of 43% of the initial amount (Figure 2). From the data obtained we concluded that compounds with a 6-carbon linker generally showed higher light stability than the derivatives with an 8-carbon linker (MB1 vs. MB3 and MB2 vs. MB4). Accordingly, MB1 and MB2 were considered the most stable compounds under long-term light exposure.

### 2.3. Effects of Natural Antioxidants, MitoBENs and Other Type of Mitochondria-Targeted Antioxidants on NHDF Cell Mass, Metabolic Viability and ATP Levels

The cytotoxicity profiles of natural parental antioxidants (Appendix A), MitoBENs (Figure 3) and MitoQ/SkQ1 (Appendix A) were next evaluated in NHDF cells by measuring cell mass, metabolic viability and intracellular ATP levels, at concentrations between 1–100 μM for 48 h.

The results with the parental compounds showed that protocatechuic and gallic acids, up to 100 μM, did not reduce cell mass (Appendix A), metabolic activity (Appendix A) and intracellular ATP levels (Appendix A) in NHDF cells. Aditionally, protocatechuic acid increased the cell mass at concentrations higher than 12.5 μM (Appendix A). On the other hand, all MitoBENs presented a concentration-dependent cytotoxicity, as a decrease in cell viability at all end-points was observed. These results suggest that all MitoBENs presented a higher cytotoxicity than their respective natural antioxidants.

Relative to catecholic compounds (MB1 and MB3), statistical analysis showed that MB1 significantly decreased the cell mass for concentrations higher than 25 μM (Figure 3A). Significant alterations in metabolic activity and ATP production were only observed when MB1 was tested at 100 μM (Figure 3B,C). MB3 caused a significant depletion in cell mass for concentrations above 6.3 μM, and in metabolic viability and intracellular ATP levels for concentrations higher than 12.5 μM (Figure 3B,C).

Relative to pyrogallol compounds (MB2 and MB4), it was observed that the treatment of cells with MB2 only showed a decrease in cell mass at 50 μM (81%) and of metabolic activity (84%) and ATP production (52%) at 100 μM (Figure 3D–F), suggesting a higher safety profile than MB1 and MB3. MB4 caused a slight decrease in cell mass (76%), metabolic activity (86%), and ATP production (90%) at 25 μM, although the difference was not statistically significant for concentrations lower than 50 μM. At 50 and 100 μM MB4 significantly decreased the three parameters when compared with control cells (Figure 3D–F). Altogether, the results also suggest that catecholic compounds (MB1 and MB3) are more cytotoxic than pyrogallol compounds (MB2 and MB4). MB3, in particular, showed the highest cytotoxic profile. Based on the viability results, we further determined MitoBEN’s genotoxic and mitochondrial toxicity profiles at the highest concentrations, i.e those that showed less than 30% reduction in cell viability.

Our data also showed a decrease in cell mass (Appendix A), metabolic activity (Appendix A) and intracellular ATP levels (Appendix A) in NHDF cells treated with MitoQ and SkQ1 at concentrations above 1 μM. These results suggested that all MitoBENs were less toxic than both mitochondria-targeted antioxidants in NHDF cells.

### 2.4. Genotoxic Effects of MitoBENs on NHDF Cells

The genotoxic profile of MitoBENs in NHDF cells was assessed using the Comet assay. Different parameters, like Comet Height, Comet Length, Tail DNA %, Tail Length, and Tail Moment were quantified using Cell Profiler software (Figure 4). The data was acquired after 48 h treatment with MB1, MB2, and MB4 at 25 μM and MB3 at 6.3 μM. The results showed that at the concentrations tested, the catecholic (MB1 and MB3) and pyrogallol compounds (MB2 and MB4) did not induce alterations in Comet Height (Figure 4A), Tail DNA % (Figure 4C), and Tail Length (Figure 4D), compared to untreated cells. However, MB1 at 25 μM significantly reduced Comet Length (Figure 4B), while MB2 significantly reduced the Tail Moment at 25 μM, compared with control cells (Figure 4E), suggesting that MB1 and MB2 could reduce basal DNA damage.

### 2.5. Mitochondrial Toxicity Profiles of MitoBENs on NHDF Cells

The mitochondrial toxicity profile of MitoBENs in NHDF cells was evaluated by measuring oxygen consumption rates (OCRs) and extracellular acidification rates (ECARs) (Figure 5), at the same concentrations of the Comet assay, using an Agilent-Seahorse XFe96 analyzer. The data showed that all MitoBENs significantly altered OCR related-parameters at the concentrations tested (Figure 5A). In fact, the catecholic (MB1 and MB3) and pyrogallol (MB2 and MB4) compounds significantly decreased the OCR related to ATP production (Figure 5C), basal respiration (Figure 5D), maximal respiration (Figure 5F), spare respiratory capacity (Figure 5H), and stressed OCR (Figure 5K) at different magnitudes. 

We also observed that MB3 increased the proton leak-based OCR and decreased the stressed ECAR (Figure 5E,J), while MB4 increased the proton leak-based OCR and the basal ECAR (Figure 5E,I), most likely as a consequence of membrane-related effects. MB1, MB3 and MB4 reduced the non-mitochondrial respiration of NHDF cells (Figure 5G). Overall, these results were in accordance with the data assessed in cytotoxic profiles evaluation, suggesting that mitochondrial effects likely cause the potential toxicity of MitoBENs in NHDF cells with MB1, MB2, and MB4 at 25 μM and MB3 at 6.3 μM. The catecholic (MB1 and MB3) compounds affected more spare respiratory capacity and stressed OCR than pyrogallol (MB2 and MB4) compounds. Even though MB2 and MB4 caused a decrease in OCR-related parameters, these effects did not result in significant cytotoxic effects to NHDF cells. Taking into account the data obtained, we selected the pyrogallol compounds MB2 and MB4 for subsequent evaluation of their antioxidant profile in the same type of cells.

### 2.6. Antioxidant Protection by MitoBENs on NHDF Cells

The antioxidant profiles of MB2 and MB4 were evaluated in NHDF cells using H_2_O_2_ as an oxidative stressor and by measuring the cell mass and metabolic activity as end-points (Figure 6).

In general, the pre-incubation of NHDF cells with H_2_O_2_ at 1250, 2500, and 5000 μM for 3 h led to a reduction of NHDF cellular mass (Figure 6A,B) and metabolic activity (Figure 6C,D). A 58–75% reduction of cell mass and 42–46% of metabolic activity was observed in NHDF cells treated with H_2_O_2_ at 1250, 2500 μM, compared to untreated cells. The treatment with 5000 μM of H_2_O_2_ reduced cell viability, with a drop to 88 and 56% of the metabolic activity (Figure 6C,D) and cell mass (Figure 6A,B), respectively, compared to untreated cells. Afterwards, NHDF cells were pre-treated with MB2 and MB4 (1.0, 3.2, 6.3, 12.5 and 25.0 μM) for 48 h and then further incubated with H_2_O_2_ (1250, 2500, and 5000 μM) for 3 h.

The sulphorodamine B (SRB) data showed an increase of cell mass with pre-treatment with MB2 (6.3–25.0 μM) and MB4 (12.5 μM) when the concentration of H_2_O_2_ was 1250 μM, (Figure 6A,B). Noteworthy in regard to the highest H_2_O_2_ concentration tested (5000 μM), only MB2 (6.3 and 25 μM) significantly protected the cells (*p* < 0.05). As shown in Figure 6C,D, protective effects on metabolic activity were observed when cells were pre-treated with MB2 (6.3–25.0 μM) and exposed to 1250 or 5000 μM H_2_O_2_. Despite a non-significant decrease in cell mass (~24%) and metabolic activity (~14%) observed with the pre-treatment of 25 μM MB4, a significant protection effect was detected in the reduction in metabolic activity induced by H_2_O_2_ at 5000 μM. Our results suggested that MB2 prevented oxidative stress-induced cell damage more efficiently than MB4, reaching significant protection in a range of concentrations between 6.3–25 μM.

### 2.7. Transcriptional Effects of MB2 on 3D EpidermFT Skin Model

As MB2 showed the lowest cytotoxic profile and more robust protective effects against oxidative stress-induced damage in NHDF cells, we next investigated its safety profile at 6.3 and 12.5 μM on the 3D EpidermFT skin model. The EpiDermFT system exhibits morphological characteristics similar to those observed in human skin, considering the contribution of skin fibroblasts and keratinocytes metabolism, secretion, and crosstalk [41]. The EpiDermFT model consists of a multilayer human dermis and epidermis composed of normal human epidermal keratinocytes and human dermal fibroblasts and is frequently used for assessing the toxicity of cosmetics and topical agents in human skin, among others [42]. The beneficial effects of MB2 on 3D EpidermFT transcripts related to mitochondrial remodeling, inflammation, and senescence were also determined.

#### 2.7.1. Effects on Viability of 3D Skin Model

The cytotoxicity of MB2 on 3D EpidermFT skin model composed of normal human epidermal keratinocytes and human dermal fibroblasts was measured using the MTT viability assay as the end-point. Our results demonstrated that MB2, at 6.3 and 12.5 μM, did not decrease the metabolic activity of the 3D EpidermFT model after a 48 h treatment (Figure 7A).

#### 2.7.2. Effects on Transcripts Related to Mitochondrial Remodeling, Inflammation, and Senescence

The 3D skin model was treated with 6.3 and 12.5 μM MB2 for a period of 48 h and the following transcripts related to different aspects of cell function were analyzed: (i) senescence: COL1A1, COL3A1, ELN, MMP1/3/9, TIMP1, GLB1; (ii) mitochondrial function: TFAM, CS, NRF1, CYCS, GABPA, PRKAA1, PPARGC1A, PINK1, SIRT1/2/3; (iii) antioxidant defences: NQO1, HMOX-1, SOD1, SOD2, GSS, NFE2L2; (iv) inflammation: TNF, IL1B, IL6, CXCL8, NFKB1, HIF1A, PPARG; (v) autophagy, senescence and cell death: BAX, TP53, BCL2, BECN1, CDKN1A, CDKN2A, LAMP2, SQSTM1, PARK2, MAP1LC3B, MTOR, LMNB1, PUM1 (Table 1). Our results showed a decrease in transcripts for MMP3 (Figure 7B), PRKAA1 (Figure 7C) and HIF1A (Figure 7D) genes in the 3D model treated with 6.3 μM MB2, compared to the untreated model. At 12.5 μM, MB2 decreased the transcription of MMP3 (Figure 7B), TFAM, PRKAA1 (Figure 7C), HIF1A, PPARG (Figure 7D) and LAMP2 genes (Figure 7E), compared to the untreated model. Overall, MB2 at 6.3 μM decreased the transcripts related to senescence by around 10 to 30%, while at 12.5 μM, MB2 also reduced the transcripts related to mitochondrial biogenesis and autophagy/senescence/cell death by ~10 and 20%, respectively.

The 3D skin model was treated with 6.3 and 12.5 μM of MB2 for 48 h. The metabolic activity was evaluated using the MTT assay and the following gene transcripts were evaluated using the Fluidigm Delta gene assay and normalized delta cycle threshold (Ct) by the reference gene *HPRT1*: *CDKN1A*, *CDKN2A*, *COL1A1*, *COL3A1*, *CS*, *ELN*, *TBP*, *BCL2*, *PUM1*, *TP53*, *PARK2*, *GABPA*, *TFAM*, *SOD1*, *NRF1*, *HIF1A*, *YWHAZ*, *PPARGC1A*, *SIRT3*, *SIRT1*, *SQSTM1*, *PRKAA1*, *PINK1*, *LAMP2*, *BAX*, *MAP1LC3A*, *CYCS*, *SOD2*, *BECN1*, *HMOX1*, *NQO1*, *MTOR*, *NFE2L2*, *IL1B*, *GSS*, *CXCL8*, *LMNB1*, *MMP1*, *MMP3*, *MMP9*, *NFKB1*, *SIRT2*, *TIMP1*, and *TNF*. Data from the metabolic activity are expressed as the mean ± SE of four independent experiments, while results are expressed as a percentage of the control (untreated cells). Results from transcript levels are expressed in delta Ct and represent the mean ± SE of four independent experiments. Statistical significance between the control group and groups treated was assessed using a *t*-test. *** *p* < 0.001, ** *p* < 0.01 and * *p* < 0.05 compared to the respective CTL (vehicle-treated cells).

A PCA analysis was then performed to select the most important transcripts that could separate the different experimental conditions. The results of the PCA analysis showed a perfect separation between different conditions: MB2-treated 3D skin model (at 6.3 and 12.5 μM) and untreated 3D skin model, based on the top 11 non-redundant variables (MMP3, PRKAA1, MTT_values, TNF, TFAM, CDKN2A, CYCS, TIMP1, TP53, PPARGC1A, HIF1A). Untreated 3D skin models presented a higher expression of transcripts related to senescene (MMP1 and MMP3), mitochondrial function (PPARGC1A and PRKAA1) and inflammation (HIF1A). 3D skin models treated with MB2 at 6.3 μM showed higher viability compared to untreated models, while 3D skin models treated with MB2 at 12.5 μM presented a higher expression of TNF and TIMP1 compared to 3D skin models treated with 6.3 μM of MB2 (Figure 8A,B). In agreement, the clustering analysis showed that untreated 3D skin models presented a higher expression of MMP3 transcripts relative to 3D skin models treated with 12.5 μM of MB2. On the other hand, 3D skin models treated with 6.3 μM of MB2 presented a higher expression of CYCs relative to 3D skin models treated with 12.5 μM of MB2 (Figure 8C). In sum, these results suggested that the treatment with MB2 promoted a general reduction of senescence markers in the 3D skin model, without affecting viability.

### 2.8. Mutagenic Effect of MB2

A critical safety requirement for human application of any compound is its non-mutagenicity. To confirm this issue, we evaluated the mutagenic potential of MB2 using the Ames assay. A dose-response study was initially performed to evaluate the cytotoxicity of MB2 in *S. typhimurium* TA100 strain and to select the maximal non-cytotoxic concentration. Our results showed that MB2 presented cytotoxicity in *S. typhimurium* TA100 only at a 5 mg/plate concentration, showing a ≥50% decrease in the number of revertant colonies compared to control (with the equivalent volume of DMSO used at 5 mg/plate) (Appendix A). Based on the cytotoxic results, the Ames reversion assay was carried out with MB2 in a range of concentrations of 0.02–1.67 mg/plate, using direct incorporation and the pre-incubation procedures with and without the S9 system. None of the concentrations assayed led to an increase in the number of revertant colonies, following the direct incorporation or pre-incubation methods, either in the presence or absence of metabolic activation in *S. typhimurium* TA98, TA100, TA1535, TA1537 and *E. coli* WP2(pKM101) strains (Figure 9, Appendix A), suggesting that MB2 was considered non mutagenic/non pro-mutagenic, under the experimental conditions assayed.

## 3. Discussion

In this work, our objective was to investigate whether MitoBENs could be potential active ingredients for topical skin products by delaying skin cell senescence. We initially evaluated the chemical stability, cytotoxicity, genotoxicity and mitochondrial toxicity of the four compounds (MB1 to MB4) and the antioxidant protective capacity of the compounds with the lowest cytotoxic profile. We next evaluated the effects of the most promising compound on transcripts related to senescence (our primary objective), as well as with other events involved in skin aging, including mitochondrial remodeling and inflammation, and also determined its mutagenic potential.

Considering that increased OS contributes to skin senescence [3,43], it is expected that skin anti-ageing products containing antioxidants would reduce the harmful effects of free radicals and minimize the consequences of skin aging. However, limitations exist for most conventional antioxidants [21]. Moreover, if antioxidants do not directly target mitochondria, their protective efficacy in that organelle is compromised [20,21]. For this reason, targeting mitochondria with antioxidants can be looked as a rational strategy to delay skin ageing. An approach to target antioxidants to mitochondria has been explored by functionalizing antioxidant cores with a lipophilic cation, such as TPP^+^. The conjugation with TPP^+^ led to a 5–10-fold higher accumulation of the active molecule in the cytoplasm, with an additional 100- or 1000-fold accumulation in mitochondria [44,45]. The potential for skin-related applications is an opportunity to study the natural-based antioxidants already reported by our group [28,35] as agents to prevent OS-induced senescence.

We initially evaluated the chemical stability of MitoBENs after a 7-day exposure to indoor light conditions to ensure that the results presented in this work were not due to the degradation of the compounds. In fact, the degradation of the phenolic compounds can occur during multiple stages, including pre-treatment, processing, long-term storage, and, especially, when exposed to drastic temperature variations, oxygen, pH, and light [46]. Even at 4 °C, the degradation of phenolic compounds under light-exposed storage conditions has already been reported, a process probably related to oxidation and polymerisation reactions [47]. As the same type of oxidative reactions can occur at 25 °C, the evaluation of the chemical stability of MitoBENs in different light exposure conditions was considered relevant. In general, MitoBENs showed great stability under light exposure, except for MB4, which presented a degradation of around 40% of compound after 7 days (Figure 2). The data suggested that compounds with a 6-carbon linker showed higher light stability than the corresponding 8-carbon linker derivatives (MB1 vs. MB3 and MB2 vs. MB4). No significant chemical stability alterations were observed comparing catecholic and pyrogallol-based systems, which allowed us to conclude that MB1 and MB2 were the most stable MitoBENs in long-term light exposure conditions.

The relationship between MitoBENs structure and their toxicity on NHDF cells was subsequently studied and compared with their respective parental antioxidants and MitoQ and SkQ1, two mitochondria-targeted antioxidants with current cosmetic applications [37,38]. Cytotoxicity of MitoBENs (1–100 μM) on NHDF cells was determined by measuring cell mass, metabolic activity, and intracellular ATP levels after a 48 h incubation. All MitoBENs showed concentration-dependent toxicity effects, translated into a significant decrease in cell viability when all compounds were tested at 100 μM. In particular, MB3 showed the highest cytotoxicity, causing a depletion of metabolic viability and intracellular ATP levels (above 12.5 μM), and cell mass (above 6.3 μM) (Figure 3). From the analysis of the cell viability data (see results Section 2.3) a cytotoxic ranking was established: MB3 < MB4 ≃ MB1 < MB2. This data complemented previous observations reported in the literature [33,48]. These previous works suggested that catecholic-based compounds display higher cytotoxicity when compared to pyrogallol based compounds and that longer alkyl chains are related with a high cytotoxic profile when compared to the shorter ones, mainly due to an increment of lipophilicity which can increase anchorage of the TPP^+^ derivatives into the mitochondrial internal membrane [33]. MB1 and MB2 were the less toxic compounds on NHDF cells when compared with the corresponding 8-carbon linker analogs (MB3 and MB4). By comparing MB1 and MB2 structures, we concluded that the presence of a third hydroxyl group in the aromatic ring increased the polarity of MB2, which lowered the cytotoxic profile. Comparing the cytotoxic profile of MitoBENs with their respective natural antioxidants, it was evident that all MitoBENs presented a slightly higher toxicity than their natural precursors (Appendix A), as a logical consequence of the inherent toxicity of TPP cations, as previously described in the literature [33]. However, it should be noted that all of them presented a much lower toxicity than MitoQ and SkQ1 (Appendix A), which are two of the most known mitochondria-targeted antioxidants [29,32], and which are already present in commercial cosmetic products.

Complementary to the cytotoxic profiles, we evaluated, for the first time, the genotoxic effects of MitoBENs. The comet assay data, a sensitive and widely used method for studying DNA damage [49] under basal conditions [50] or after different pharmacological treatments [51,52,53], showed that none of the MitoBENs under study induced genotoxic effects, as observed by Tail DNA % and Tail Length data (Figure 4C,D). We next investigated, for the first time, the potential mitochondrial toxicity of these compounds by measuring OCR and ECAR parameters using the Seahorse technique. For all MitoBENs, we observed a reduction in maximal and stressed OCR (Figure 5) that suggested the inhibition of non-phosphorylative respiration at concentrations lower than those that decrease cell viability (Figure 3), which could be caused by inhibition of the respiratory chain or alterations in the lipid membrane. Based on the impairment of non-phosphorylative respiration the mitochondrial toxicity ranking was MB3 < MB1 < MB4 < MB2. Based on these results, since MB3 and MB1 were the two compounds with higher cytotoxicity in NHDF cells, the following work was only performed with the pyrogallol–based compounds (MB2 and MB4). Our results showed that MB2 and MB4 showed antioxidant capacity in NHDF cells, as observed by the prevention of the decrease of cell mass and metabolic activity caused by H_2_O_2_, with MB2 being more potent than MB4 (Figure 6). Considering that MB2 and MB4 are both pyrogallic derivatives and they differ in the sizes of their carbon linkers, it is possible that the type of spacer can influence the antioxidant protection effect of these compounds, complementing previous information [48].

The data obtained using the more complex 3D EpidermFT skin model, which exhibits morphological characteristics similar to those observed in human skin [41], supported the potential use of MB2 as a skin anti-ageing molecule, targeting processes of cell senescence. At concentrations that did not affect the viability of the 3D EpidermFT skin model, MB2 decreased transcripts for important senescence markers [54], namely MMP3, HIF1A and PRKAA1 (Figure 7B–D and Figure 8A,B). The fact that MB2 decreased transcripts for MMP3 could be an indicator of an additional protection mechanism that could delay the senescence process, similar to that described for Curcuma mangga extract (composed by gallic acid, catechin, epicatechin, epigallocatechin, epigallocatechin gallate, and gallocatechin gallate) that reduces protein expression of MMP1, MMP3 and MMP9 and protects fibroblasts from collagen degradation induced by OS [55]. Additionally, as MB2 can induce the regulation of HIFA gene transcription, it may also have beneficial effects on the inflammatory process, since HIF-1α increases the production of different proinflammatory cytokines, including TNF-α, IL-1β, IL-6, and IL-8 [56]. This effect may also contribute to mitochondrial remodeling, preventing cellular senescence. In fact, HIF-1α induces a deficit in mitochondrial biogenesis by reducing the transcription of TFAM, required for replication, transcription, and maintenance of mitochondrial biogenesis, and promoting the inhibition of the PGC-1β activity impairs energy-dependent cellular processes, including cell and tissue repair [57]. Alternatively, the reduction of the transcription of PRKAA1 gene promoted by MB2 may also lead to decreased senescence, since PRKAA1 is a gene that encodes the AMPK, described as causing premature fibroblast senescence [58]. These results suggest that MB2 can delay the skin ageing process inducing different and complementary mechanisms of action related to senescence inhibition and mitochondrial remodeling. However, these mechanisms should be clarified in the future by measuring protein levels and enzymatic activities of the different markers involved in these multiple pathways. 

The absence of mutagenic and pro-mutagenic effects of MB2 observed in the AMES test, performed following OECD Guideline 471 for the Testing of Chemicals (Figure 9, Appendix A), is a critical element for the safety assessment of that molecule [59].

In conclusion, our results suggest that the pyrogallol based compounds (MB2 and MB4) showed a better safety profile in vitro than catecholic based compounds (MB3 and MB1). The pyrogallol derivative, MB2, containing a 6-carbon spacer, showed a good biological safety profile, chemical stability, antioxidant activity, and no mutagenicity. These characteristics, combined with the effects on transcripts related with skin senescence, inflammation and mitochondrial remodeling, make MB2 a good candidate for further development as an active ingredient for anti-skin ageing formulations. 

## 4. Materials and Methods

### 4.1. Reagents

All reagents and solvents for chemical synthesis and biochemical assays were used without any further purification and purchased from Sigma-Aldrich (St. Louis, MO, USA), Carlo Erba Reactifs (Lisbon, Portugal), and Panreac (Barcelona, Spain). Fetal bovine serum (FBS) (10270106), penicillin streptomycin (15140122), tetramethylrhodamine, methyl ester (TMRM) (T668) and Hoechst 33,342 (H1399), and trihydrochloride, trihydrate (H1399) were acquired from Gibco-Invitrogen (Waltham, MA, USA). Dulbecco’s Modified Eagle’s Medium (DMEM-D5030), L-glutamine (G3126), sodium bicarbonate (S6014), sodium pyruvate (P5280), 2-[4-(2-hydroxyethyl)piperazin-1-yl]ethanesulfonic acid (HEPES) (H3375), sulforhodamine B sodium salt (S9012), hydrochloric acid 37% (30721), hydrogen peroxide (107210-1000), resazurin sodium salt (R7017), dimethyl sulfoxide (DMSO) (34869), carbonyl cyanide 4-(trifluoromethoxy)phenylhydrazone (FCCP) (C2920), rotenone (R8875), antimycin A (A8674), triton X-100 (T9284-0100), ethylenediaminetetraacetic acid disodium salt dihydrate (EDTA) (ED2SS-0100), protocatechuic acid (37580) and gallic acid (G7384) were acquired from Merck Life Science S.L.U (Darmstadt, Germany). Oligomycin (495455) was purchased from Calbiochem (Darmstadt, Germany). From VWR (Radnor, PA, USA), we obtained d-glucose (0188), agarose universal (732-2789), and low-melting agarose (H26417.14). Tris base was obtained from ChemCruz (sc-3715) (Heidelberg, Germany). Glacial acetic acid (A/0400/PB15), methanol (M/4056/17), ethanol absolute (E/0650DF/C17) and sodium hydroxide (P/5640/60) were from Fisher Chemical (Pittsburgh, PA, USA).

### 4.2. Chemistry

#### 4.2.1. General Conditions

The progression of the chemical reactions was followed by thin-layer chromatography (TLC) and the purification of the compounds was performed by flash column chromatography using silica gel 60 0.040–0.063 mm (Carlo-Erba Reactifs, Val-de-Reuil, France) or cellulose 0.01–0.10 mm (Sigma-Aldrich) as described by Oliveira et al. [39]. The compound’s structural characterization was performed by 1H and 13C nuclear magnetic resonance (NMR), electron impact mass (EI) and electrospray ionization (ESI) spectroscopies. The conditions were reported by Oliveira et al. [39].

#### 4.2.2. Synthesis of MitoBENs, MitoQ and SkQ1

The synthetic methodologies and spectroscopic characterization data (NMR, EI-MS and ESI) of the hydroxybenzoic acid-based antioxidants (MB1–MB4) were previously described by Oliveira et al. [28,39]. The synthesis of MitoQ and SkQ1 was based on the procedures reported by James et al. [60] and Korshunova et al. [61].

#### 4.2.3. Determination of MitoBENs Stability

##### Analytical Conditions

The stability of the antioxidants under indoor light conditions was evaluated with a NEXERA-i LC-2040C UHPLC apparatus (Shimadzu, Kyoto, Japan) equipped with a diode array detector and controlled by the LabSolution system (version 5.90 Shimadzu, Japan).

The reverse phase-ultra high-performance liquid chromatography (RP-UHPLC) experiments were performed on a Luna 3 µm C18 100 Å (15 cm) column at a flow rate of 1 mL/min with the chromatographic column temperature set to 20.0 °C. The injected volume was 50 μL.The elution system was composed by water containing 0.1% glacial acetic acid (eluent A) and acetonitrile (eluent B). The gradient elution conditions were the following: Time: 0–1.5 min, A = 100%; Time: 1.5–3 min, A = 100 to 0% and B = 0 to 100%; Time = 3–5 min, B = 100%; Time = 5–5.5 min, A = 0 to 100% and B = 100 to 0%; Time = 5.5–8 min, A = 100 to 0% and B = 0 to 100%. The detection wavelengths were 257 and 266 for catechol and pyrogallol-based derivatives, respectively. The results were presented as mean ± standard error (SE) of three independent experiments.

##### Assay Conditions

Briefly, compound stock solutions (4 mM) were prepared in DMSO and diluted in ultrapure water (pH 7.4) to achieve a final concentration of 100 μM. The resulting solutions were transferred to conical tubes (V = 6 mL) and led at ambient light exposure and room temperature. An aliquot of the work stock solution was taken at predetermined periods, an equivalent acetonitrile amount was added, filtered, and analyzed by UHPLC. The concentration at the different time-points (*Tx*) was calculated using calibration curves obtained under sink conditions (Appendix A). Three independent experiments were performed. The data is presented as the mean percentage of remaining antioxidant, calculated by the Equation (1), where *T*0 is the initial concentration solution (t = 0 min).
(1)$$Percentage\;of\;remaining\;antioxidant=\frac{Conc\;\left(Tx\right)}{Conc\;\left(T0\right)}\times 100$$

### 4.3. Cell culture Conditions and Treatments with Parental Antioxidants, and Mitochondria-Targeted Antioxidants

Normal human dermal fibroblasts (NHDF), with batch n° 1839 and reference 106-05a, were purchased from Cell Applications, Inc (San Diego, CA, USA). Normal human dermal fibroblasts were cultured in DMEM, D5030 supplemented with 10% (*v*/*v*) FBS, 100,000 units/L penicillin, 100 mg/L streptomycin, 44 mM sodium bicarbonate, 6 mM L-glutamine and 1 mM sodium pyruvate and 5 mM glucose. Monolayer cell cultures were maintained in 100 and 150 mm tissue culture dishes (VWR-references I734-2321 and I734-2322 respectively) at 37 °C in a humidified atmosphere with 5% CO_2_. The culture media was changed every other day, or at most every three days. Cells were subjected to trypsinization using a 0.05% trypsin-EDTA (*v*/*v*) solution when 80–90% confluence was achieved. 

Cells were used at a density of 1.5 × 10^4^ cells/cm^2^ in all experiments, except for the Seahorse experiments, in which NHDF density was 2 × 10^4^ cells/80μL/well. 

Stock solutions of parental antioxidants (protocatechuic and gallic acids), MitoBENs (MB1, MB2, MB3, MB4) and MitoQ and SkQ1 were prepared in DMSO (100 mM) and further diluted in the respective culture media at concentrations between 1–100 μM. The maximal concentration of DMSO used in the cellular treatments was 0.1% (*v*/*v*).

#### 4.3.1. Determination of Cellular Metabolic Activity and Cell Mass

The determination of metabolic viability and cell mass was performed using resazurin and sulforhodamine B (SRB), respectively, according to a previously published protocol [62]. After 48 h of treatment, the growth media was replaced by fresh media containing 10 μg/mL resazurin. The cells were incubated for 1.5 h at 37 °C in a humidified atmosphere with 5% CO_2_. The reduction of resazurin to resorufin, an indirect measurement of the number of viable/metabolically active cells [62], was measured by fluorescence (540 nm excitation, 590 nm emission) using a CLARIOstar Plus microplate reader (BMG Labtech, Ortenberg, Germany).

After determining cellular metabolic activity, the resazurin/resorufin-containing medium was removed. Cells were fixed overnight at −20 °C with 100 μL of 1% (*v*/*v*) acetic acid in methanol. After discarding the fixation solution, the plates were air-dried and incubated at 37 °C for 1 h with 70 μL/well of a 0.05% (*w*/*v*) SRB solution. After this incubation, the SRB solution was discarded, and the cells were thoroughly washed with a solution of 1% (*v*/*v*) acetic acid in ultrapure water. The plates were once again air dried, and the protein bound to SRB was dissolved in 125 μL of 10.5 mM Tris-NaOH. The SRB absorbance was measured at 510 nm using a CLARIOstar Plus microplate reader (BMG Labtech, Ortenberg, Germany). Absorbance at 620 nm was used to correct for background absorbance for each well.

#### 4.3.2. Evaluation of Intracellular ATP Levels

Intracellular ATP levels were measured using the CellTiter-Glo^®^ Luminescent Cell Viability Assay (PROMG7571, Promega (Madison, WI, USA) according to manufacturer instructions. Briefly, cells were seeded and treated with MitoBENs, as previously described in the Section 4.3, using white opaque 96-well plates (Costar-reference CORN3917). At the end of the compounds’ incubation period the growth media was replaced by 40 μL of fresh room temperature equilibrated growth media, and 40 μL of CellTiter-Glo^®^ reagent mix was added to each well to promote cell lysis and the luciferase-catalysed oxidation of luciferin to oxyluciferin. The reaction occurred for 10 min. The luminescent signal generated is proportional to the amount of ATP and measured using a CLARIOstar Plus microplate reader (BMG Labtech, Ortenberg, Germany).

#### 4.3.3. Measurement of DNA Damage

Comet assay, also called single cell gel electrophoresis (SCGE), measures DNA damage. The assay is based on the migration of denatured DNA, through an electrophoretic field [63]. In this assay, microscope slides were covered with 1% (*w*/*v*) normal agarose and left to dry on horizontal until the agarose was fully solid. Slides were placed in an oven at 50 °C to dry the agarose for 1 to 2 h and stored at room temperature (RT) in a container to protect them from dust and humidity until its use.

Cells were seeded and treated with MitoBENs (MB1, MB2, and MB4 at 25 μM and MB3 at 6.3 μM) for 48 h in a 12-well plate. For the assay, cells were trypsinized with 0.05% (*v*/*v*) of trypsin-EDTA solution. Pre-chilled cell culture medium containing FBS was added to inhibit trypsin activity. All steps were performed in the dark to minimise DNA damage during the experimental procedure. Cells were transferred to pre-chilled 15 mL conical tubes and centrifuged at 250× *g* and 4 °C during 5 min. The pellet was then resuspended in 50 μL of pre-chilled filtered PBS. Twenty μL of cells were mixed with 80 μL of 1% (*w*/*v*) low melting point agarose, and 70 μL of the cell suspension was immediately transferred to an agarose pre-coated microscope slide and covered with a slide cover. After agarose solidification, coverslips were carefully removed, and slides were covered with pre-chilled lysis solution and left for 1 h at 4 °C. Slides were then placed on the platform of the electrophoresis chamber filled with a pre-chilled alkaline unwinding solution/electrophoresis buffer. The electrophoresis was run at 25 V for 20 min in a multiSUB Choice electrophoresis system from Cleaver Scientific (Warwickshire, UK). At the end of the electrophoresis run, slices were gently removed from the electrophoresis chamber and immersed in a neutralisation solution composed of 0.4 M Tris, pH 7.5 for 5 min at 4 °C. Eighty μL of Hoechst 33,342 staining solution 1 mg/mL was added to each slide and incubated for 30 min. Images were acquired using a Zeiss AxioVert A1 microscope (Jena, Germany). The images of 100 NHDF cells were captured per sample and different parameters were calculated, including Comet Height, Comet Length, Tail Length and Tail DNA %, defined by the percentage of DNA in the tail. Tail Moment was also calculated, as the product of tail length and total DNA fraction in the tail.

#### 4.3.4. Determination of Oxygen Consumption Rate (OCR) and Extracellular Acidification Rate (ECAR)

Oxygen consumption rate and ECAR parameters were measured at 37 °C using a Seahorse XFe96 Extracellular Flux Analyzer (Agilent Scientific Instruments, Santa Clara, CA, USA). 

Cells were seeded and treated with MitoBENs, as previously described in Section 4.3, using XF96 cell culture microplates (Seahorse XFe96 FluxPak 102416-100, Agilent, Santa Clara, CA, USA). The day before the experiment, XFe96 sensor cartridges (Seahorse XFe96 FluxPak 102416-100, Agilent) were pre-hydrated overnight with 200 μL/well of calibration buffer at 37 °C in a humidified atmosphere without controlled CO_2_ levels. On the following day, the culture media was replaced with 175 μL/well of pre-warmed low buffered serum-free minimal DMEM medium, supplemented with 6 mM glutamine, 1 mM sodium pyruvate and 5 mM glucose, pH 7.4. Microplates were incubated at 37 °C for 1 h to allow the temperature and pH of the medium to reach equilibrium before the first-rate measurement. Oligomycin, FCCP, rotenone (ROT), and antimycin A (AA) were prepared in low-buffered serum-free DMEM medium, supplemented with glucose, glutamine, and sodium pyruvate, at pH 7.4. Oligomycin (2 μM) FCCP (2 μM) and ROT/AA (1 μM) were loaded into cartridge ports A, B and C, respectively: After 1 h of calibration, the 96-well calibration plate was replaced by the cell plate. Oligomycin, FCCP, and ROT/AA were then injected at sequential times by the XFe96 Analyzer into each well. The analysis of the OCR-related parameters was performed using the Agilent Seahorse XF Cell MitoStress test report generator software (version 2.6.0).

#### 4.3.5. Evaluation of the Protective Capacity of MB2 and MB4 against Oxidative Stress Inducers

The protective capacity of MB2 and MB4 against OS induced by H_2_O_2_ was evaluated after 3 h of treatment. Cells were pre-treated with MB2 and MB4 (1–25 μM) for 48 h. Then, the culture medium was replaced, and the cells were treated with H_2_O_2_ (1250, 2500, and 5000 μM) for 3 h at 37 °C. After the incubation time, cell mass and cellular metabolic activity were determined, as previously reported in the Section 4.3.1.

### 4.4. Viability and Gene Expression Assays Using the 3D Epiderm FT Skin Model

#### 4.4.1. Determination the Viability of 3D Skin Model

A 3D EpidermFT skin model composed of normal human epidermal keratinocytes and human dermal fibroblasts with the reference numbers EFT400 (kit of 24 inserts) and EFT418 (kit of 18 inserts) was purchased from MatTek Corporation (Ashland, MA, USA). The cytotoxic effect of MB2 on the 3D EpidermFT skin model was evaluated using the MTT assay (MTT-400, MatTek Corporation), as advised per manufacturer instructions for “EpiDerm Full Thickness 400 (EFT-400)”. At the end of 48 h of incubation with 6.3 and 12.5 μM MB2, EFT-400 and EFT418 inserts were washed three times with PBS and placed in a clean 6-well plate containing the MTT solution (2 mL/well.) After 3 h of incubation at 37 °C, each insert was carefully removed, immersed in an extraction solution supplied by the commercial kit MTT-400, and incubated for 2 h at room temperature in the dark. Then, 200 μL of cellular suspension was transferred from each well into 96-well plates. The optical density of the samples was analyzed at 570 nm. The background readings were determined at 650 nm and subtracted to the optical density measured at 570 nm.

#### 4.4.2. Determination of Gene Expression

After 48 h of incubation with MB2, EFT-418 inserts were washed three times with PBS, then plucked from the membrane, disrupted with forceps and deposited directly into RLT buffer, which is a lysis buffer provided from Qiagen RNeasy Mini kit (Qiagen, Düsseldorf, Germany). Scissors were used to mince the inserts containing the co-culture of cells. Once minced, the cellular suspensions were homogenised using a homogenizer apparatus Ultra Turrax IKA T10 (Staufen, Germany). Total RNA was extracted following the manufacturer’s protocols, using Qiagen RNeasy Mini kit (Qiagen, Düsseldorf, Germany), and quantified using a Nanodrop 2000 (Thermo Fisher Scientific, Waltham, MA, USA). All samples presented a A260/280 ratio between 1.8 and 2.1. Each RNA sample was diluted in water to the same concentration (16.5 ng/μL). To obtain complementary DNA (cDNA) reverse transcriptase enzyme (1 μL) [qScript cDNA super mix (Quanta BioSciences, Beverly, MA, USA)] was added to 4 μL of each RNA sample. Each cDNA sample was pre-amplified. Briefly 1.25 μL of each cDNA sample was mixed with 0.5 μL of a mix of pooled primers (500 nM final concentration each), 2.25 μL water and 1 μL of PreAmp Master Mix enzyme (Fluidigm, San Francisco, CA, USA). The primers were obtained from Fluidigm, as described in Table 2, and reconstituted to a final concentration of 100 μM in water. Thermal cycling was performed using the CFX Connect Real-Time PCR System (Biorad, Hercules, CA, USA) according to the enzyme manufacturer for 12 cycles. The samples were then treated with Exonuclease I (New England Biolabs, Hitchin, UK) to remove unincorporated primers. After Exonuclease I treatment, the samples were diluted 10× in TE buffer (10 mM Tris-HCl, 1 mM EDTA). For each sample (2.25 μL) a Pre-Mix was prepared with 2.5 μL SsoFast EvaGreen Supermix (BioRad, Hercules, CA, USA), 0.25 μL of 20× DNA binding dye sample reagent (Fluidigm, San Francisco, CA, USA). A volume of 5 μL of each sample was pipetted into the respective inlet of a Fluidigm^®^ 48.48 Gene expression IFC. For each gene assay a mix of 12 μL was individually prepared: 6 μL of 2× Assay loading reagent (Fluidigm, San Francisco, CA, USA), 5.4 μL of TE buffer, 1.2 μL from a stock of 50 μM each mixed forward and reverse primers. A volume of 5 μL of each assay was pipetted into their respective assay inlets on the chip. The assay and sample mixes were loaded with the corresponding Load mix script of the MX controller (HD Biomark Fluidigm, San Francisco, CA, USA). After loading, the chip qRT-PCR was carried out using the BioMark HDTM, according to the cycling parameters recommended by Fluidigm for 48.48 Gene expression IFC. Data were collected with Data Collection Software and analyzed using Fluidigm^®^ Real Time PCR Analysis v2.1 software. Genes with melting curves displaying more than one peak (amplification of non-specific products) were not included in the analysis. The data were normalized for the reference gene HPRT1. This gene codifies an enzyme involved in recycling guanine and inosine in the purine salvage pathway, being expressed at constant low levels in all somatic tissue [64], and frequently used as a reference gene in different studies [65,66,67].

### 4.5. Evaluation of the Mutagenic Potential

The Ames assay was performed by Vivotecnia Research (Madrid, Spain) following OECD Guideline 471 [68]. The evaluation of the cytotoxicity of MB2 (from 0.06 up to 5 mg/plate) was initially performed in *S. typhimurium* TA100 strain, with and without a metabolic activation system (S9), using the MTT assay. The mutagenic or pro-mutagenic potential of MB2 was evaluated by the number of revertant colonies upon exposure to MB2 relative to the number of spontaneously occurring revertant colonies in the controls. Then, different bacterial strains (*S. typhimurium* TA98, TA100, TA1535, TA1537 and *E. coli* WP2(pKM101)) were exposed to MB2 at 5 concentrations (0.02–1.67 mg/plate) with and without the S9 system using direct incorporation and pre-incubation procedures. In the direct incorporation procedure, the suspension of bacterial cells exposed to MB2 (in the presence or absence of S9 system) was immediately poured over a minimal agar medium plate, while in the pre-incubation procedure, this suspension was incubated for 20 min at 37 °C prior to being poured over the agar plate. As positive controls: 5 mg/plate of 2-Nitrofluorene, 0.4 mg/plate of 4-nitroquinoline-N-oxide, 45 mg/plate of 9-aminoacridine were used without metabolic activation (−S9), while 2.5 and 3.5 mg/plate of sodium azide was used with metabolic activation (+S9). At the end, all plates were incubated for 48 h at 37 °C and colonies were counted using Sorcerer colony counter (Perceptive Instruments, Haverhill, UK).

### 4.6. Statistical Analyses

Statistical analyses were accomplished using R statistical software. Planned comparisons were performed between each compound and the un-treated control or between different compounds using Welch’s *t*-test. The *p*-values were corrected using Bonferroni’s methodology and deemed statistically significant if <0.05. Two cluster analyses were performed. The first was focused on cell viability data (cell mass, metabolic activity, ATP levels, mitochondrial polarization, OCR and ECAR levels). The second cluster analysis focused on the data obtained from 3D skin model study (viability and gene expression). Both cluster analyses followed a similar workflow. Firstly, data dimensionality reduction technique was applied through a random forest algorithm, and only the variables that better separated the groups were selected for the next phase. Secondly, a principal component analysis (PCA) was performed to visualize which groups of variables allowed a better explanation of the separation of data, thus, allowing the selection of the most important biological variables. Thirdly, the k-means clusters were calculated, with k being equal to the number of treatments. The resulting means closest to the observations were plotted in a scatter plot of the two most critical biological variables, each one with samples that underwent treatment with MB2 and healthy control samples, in order to visualize the relationships between different treatment conditions.

## Figures and Tables

**Figure 1 molecules-27-06183-f001:**
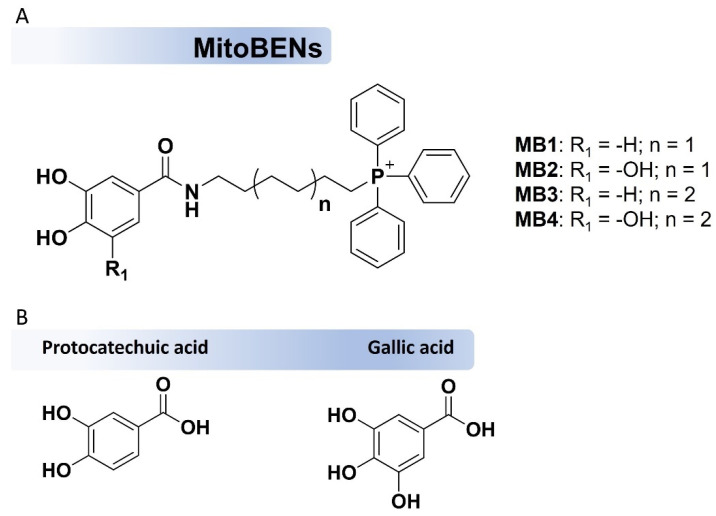
Chemical structures of hydroxybenzoic-based mitochondriotropic antioxidants MB1-MB4 (**A**) and the natural antioxidants protocatechuic and gallic acids (**B**).

**Figure 2 molecules-27-06183-f002:**
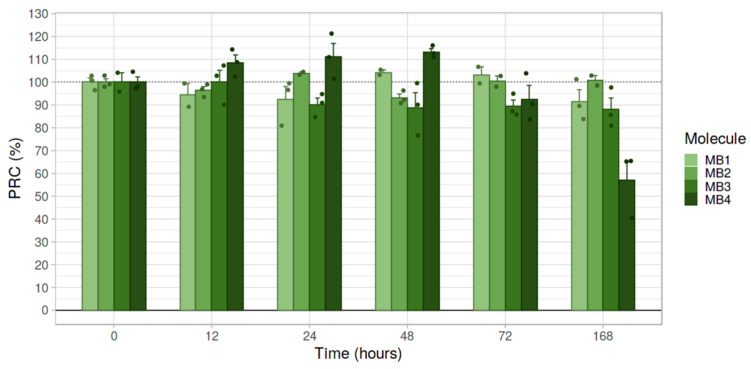
Evaluation of the stability of hydroxybenzoic-based mitochondriotropic antioxidants MB1-MB4 under indoor light conditions at room temperature for 168 hours (7 days), by measuring the percentage of remaining compound (PRC%) by UHPLC. Stock solutions of MB1-MB4 were prepared in DMSO at 4 mM and diluted in ultrapure water to achieve a final concentration of 100 μM. Data are the mean ± standard error (SE) of three independent experiments. Results are expressed as the percentage of concentration in predetermined time-points compared with initial concentration solutions (t = 0 min).

**Figure 3 molecules-27-06183-f003:**
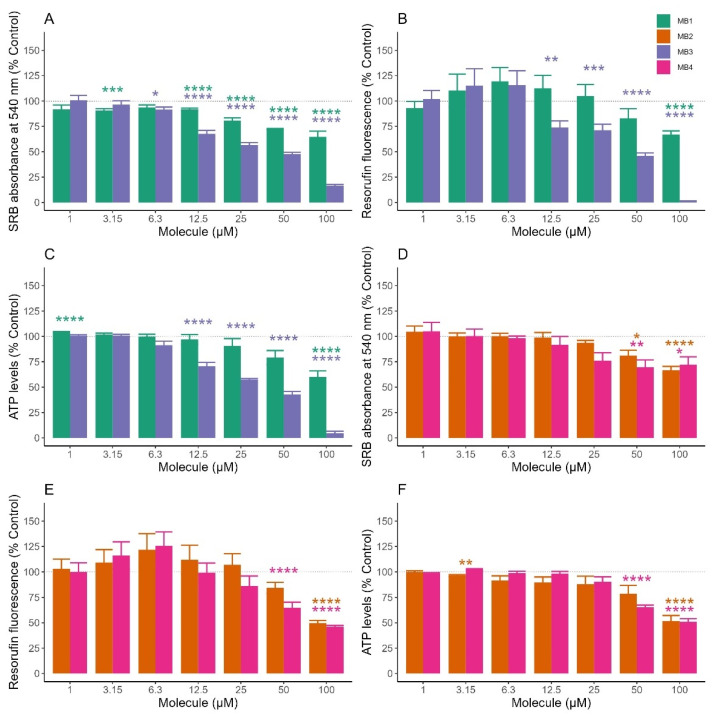
Effects of hydroxybenzoic-based mitochondriotropic antioxidants MB1-MB4 on cell mass (**A**,**D**), metabolic activity (**B**,**E**), and intracellular ATP (**C**,**F**) of normal human dermal fibroblasts (NHDF). Normal human dermal fibroblasts were treated with increasing concentrations of the different molecules for 48 h. Cellular mass, metabolic activity, and ATP levels were evaluated using sulforhodamine B (SRB) assay, resazurin reduction assay, and CellTiter-Glo Luminescent Cell Viability Assay, respectively. Data are the mean ± SE of four independent experiments. The results are expressed as a percentage of the control. Statistically significant differences between control (CTL) and treated groups were evaluated using a *t*-test. **** *p* < 0.0001, *** *p* < 0.001, ** *p* < 0.01 and * *p* < 0.05 compared to the respective control (CTL, vehicle-treated cells), with the colours green, orange, blue and pink to correspond to each compound MB1, MB2, MB3 and MB4, respectively.

**Figure 4 molecules-27-06183-f004:**
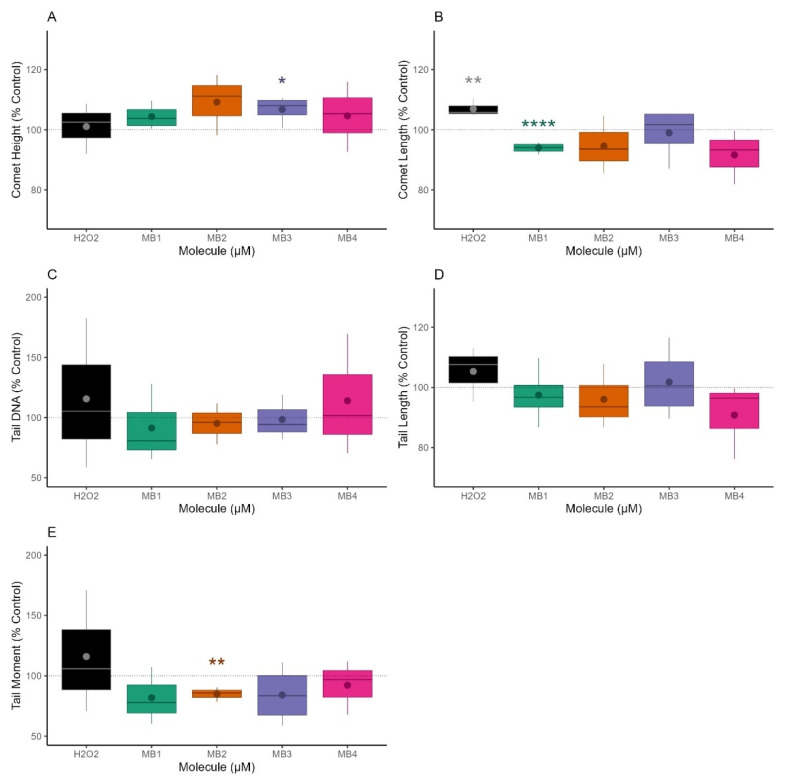
Effects of hydroxybenzoic-based mitochondriotropic antioxidants MB1-MB4 on DNA damage. Normal Human Dermal Fibroblasts (NHDF) were treated with MB1, MB2, and MB4 at 25 μM and MB3 at 6.3 μM for a period of 48 h. DNA damage was assessed using the comet assay. H_2_O_2_ at 500 μM was used as a positive control. The different parameters of Comet Height (**A**), Comet Length (**B**), Tail DNA % (**C**), Tail Length (**D**) and Tail Moment (**E**) in NHDF were quantified using Cell Profiler software. Results are expressed as a function of control (CTL = 100%) and represent the interquartile range (Q1–Q3), together with the (⦁) mean and (⎯) median of three independent experiments. Data outside the Q1–Q3 range are represented as data outliers (*). Statistically significant differences between control (CTL) and treated groups were evaluated using a *t*-test. **** *p* < 0.0001, ** *p* < 0.01 and * *p* < 0.05 compared to CTL (vehicle-treated cells).

**Figure 5 molecules-27-06183-f005:**
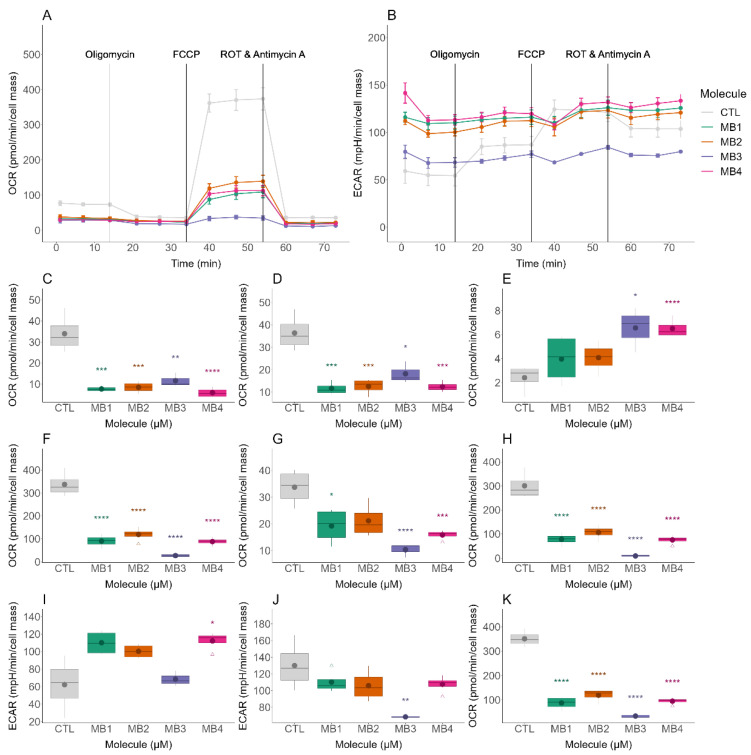
Effects of hydroxybenzoic-based mitochondriotropic antioxidants MB1-MB4 on oxygen consumption rate (OCR) and extracellular acidification rate (ECAR). OCR- and ECAR-associated parameters were assessed with a Seahorse XFe96 Extracellular Flux Analyzer. OCRs (**A**) and ECARs (**B**) were assessed in normal human dermal fibroblast (NHDF) cells treated with MB1, MB2, and MB4 at 25 μM and MB3 at 6.3 μM for 48 h. Several OCR parameters were evaluated: ATP production-linked OCR (**C**), basal respiration (**D**), proton leak-based OCR (**E**), maximal respiration (**F**), non-mitochondrial respiration (**G**) and spare respiratory capacity (**H**). ECAR parameters were also evaluated, including basal ECAR (**I**), stressed ECAR (**J**), and stressed OCR (**K**). Data are the mean ± SE of three independent experiments and the results are expressed in the interquartile range (Q1–Q3) together with the (−) median. Statistically significant differences between control (CTL) and treated groups were evaluated using a *t*-test. **** *p* < 0.0001, *** *p* < 0.001, ** *p* < 0.01 and * *p* < 0.05 compared to the respective CTL (vehicle-treated cells).

**Figure 6 molecules-27-06183-f006:**
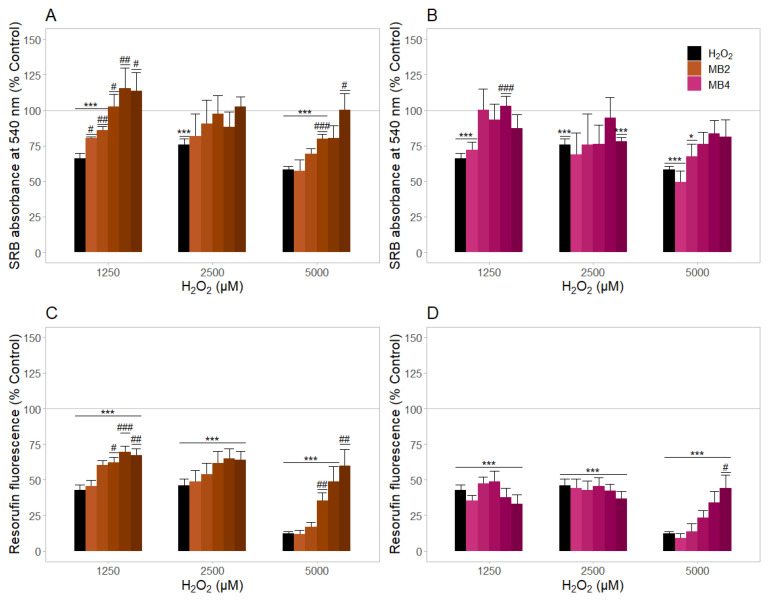
Protective effects of MB2 and MB4 against oxidative stress-induced cytotoxicity on cell mass (**A**,**B**) and metabolic activity (**C**,**D**) of normal human dermal fibroblasts (NHDFs). Normal human dermal fibroblasts were treated with MB2 and MB4 at concentrations between 1.0, 3.2, 6.3, 12.5 and 25.0 μM for 48 h. Three hours before measuring cellular mass and metabolic activity, cells were treated with increasing concentrations of H_2_O_2_ (1250, 2500 and 5000 μM). Cellular mass and metabolic activity were evaluated using the sulforhodamine B (SRB) assay and resazurin reduction assay, respectively. Data are the mean ± SE of four independent experiments. Results are expressed as percentage of the control (antioxidant vehicle-treated cells). Statistically significant differences between control and treated groups or between H_2_O_2_ (no antioxidants) and treated groups were evaluated using a *t*-test. *** *p* < 0.001 and * *p* < 0.05 compared to the respective control without antioxidant. ### *p* < 0.001, ## *p* < 0.01 and # *p* < 0.05 compared to the respective control treated with H_2_O_2_ without antioxidant.

**Figure 7 molecules-27-06183-f007:**
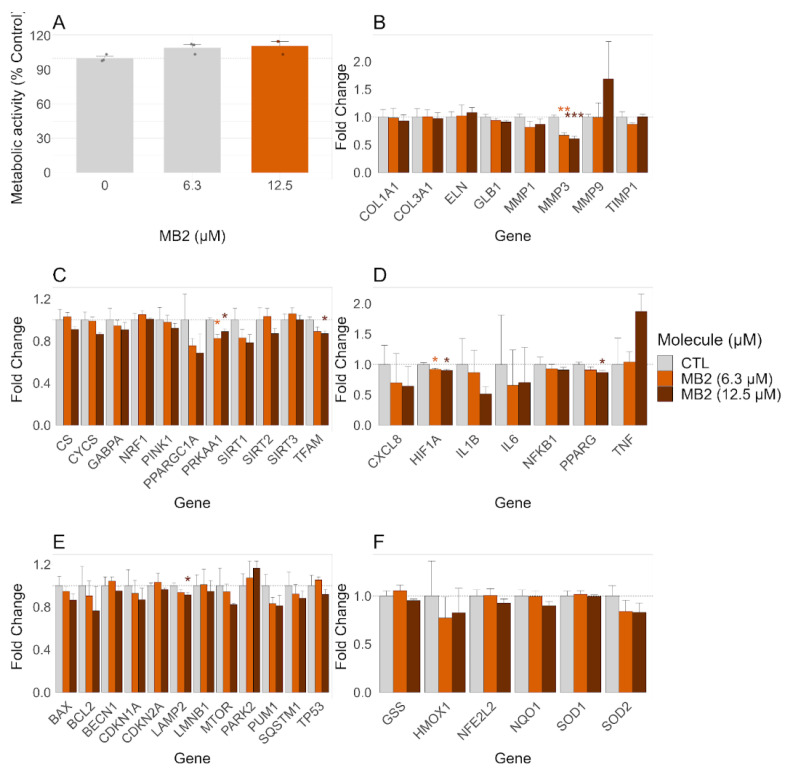
Effect of MB2 on the metabolic activity (**A**) and on the transcripts relevant to mitochondrial biogenesis, for the modulation of mitochondrial metabolism, and measurement of anti-inflammatory mediators (**B**–**F**) in the 3D EpidermFT skin model.

**Figure 8 molecules-27-06183-f008:**
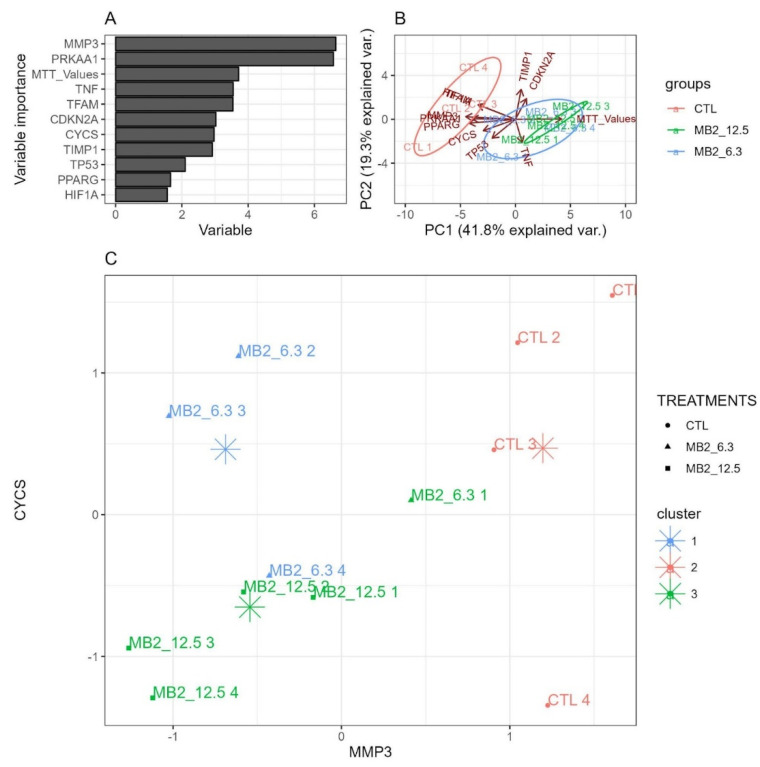
Information gain of different transcripts and MTT values that resulted from a random forest algorithm for condition prediction. Panel (**A**) shows the top 11 non-redundant variables that best split the data. Panel (**B**) shows a PCA data reduction on the top 11 variables. Panel (**C**) shows a k-means cluster analysis of the 6 least correlated variables that best split the data (PCA) into 3 clusters.

**Figure 9 molecules-27-06183-f009:**
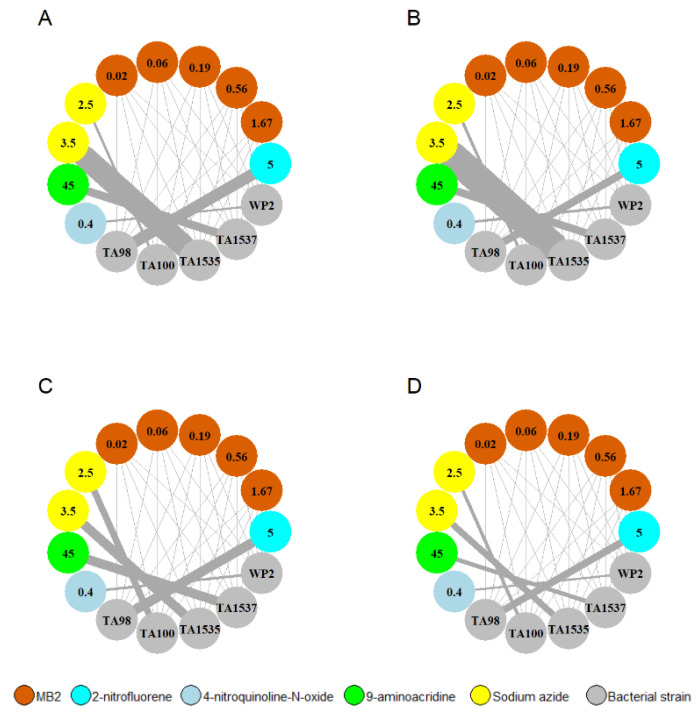
Genotoxic effects of MB2 using Ames assay. Different strains of *S. typhimurium* (TA98 and TA100), *E. Coli* (WP2) and *S. typhimurium* (TA1535 and TA1537) were tested by: (**A**) direct incorporation procedure without metabolic activation; (**B**) pre-incubation procedure without metabolic activation; (**C**) direct incorporation procedure with metabolic activation; (**D**) pre-incubation procedure with metabolic activation. The representation reflects the amount (mg/plate) of each compound (MB2, 2-nitrofluorene, 4-nitroquinoline-N-oxide, 9-aminoacridine, and sodium azide) in each sphere, while the connection lines width are proportionally direct to the ratio of revertant colonies per exposed plate (R). Ratio value (R) was obtained by comparison between the triplicate count mean of DMSO control solvent and the triplicate count mean of each concentration of compound tested.

**Table 1 molecules-27-06183-t001:** Name of transcripts and respective functional involvement.

Name of Transcripts	Functional Involvement
*Collagen Type I Alpha 1 Chain (COL1A1)*	Senescence
*Collagen Type III Alpha 1 Chain (COL3A1)*	Senescence
*Elastin (ELN)*	Senescence
*Matrix Metallopeptidase 1*/*3*/*9 (MMP1*/*3*/*9)*	Senescence
*Metallopeptidase Inhibitor 1 (TIMP1)*	Senescence
*Galactosidase Beta 1 (GLB1)*	Senescence
*Transcription factor A, mitochondrial (TFAM)*	Mitochondrial function/biogenesis
*Citrate synthase (CS)*	Mitochondrial function
*Nuclear Respiratory Factor 1 (NRF1)*	Mitochondrial function/biogenesis
*Cytochrome C (CYCS)*	Mitochondrial function
*GA Binding Protein Transcription Factor* *Subunit Alpha (GABPA)*	Mitochondrial function
*Protein Kinase AMP-Activated Catalytic* *Subunit Alpha 1 (PRKAA1)*	Mitochondrial function
*Peroxisome proliferator-activated receptor gamma coactivator 1-alpha (PPARGC1A)*	Mitochondrial function/biogenesis
*PTEN-induced kinase 1 (PINK1)*	Mitochondrial function
*Sirtuin 1*/*2*/*3 (SIRT1*/*2*/*3)*	Mitochondrial function
*NAD(P)H Quinone Dehydrogenase 1 (NQO1)*	Antioxidant defenses
*Heme Oxygenase 1 (HMOX-1)*	Antioxidant defenses
*Superoxide Dismutase 1 (SOD1)*	Antioxidant defenses
*Superoxide Dismutase 2 (SOD2)*	Antioxidant defenses
*Glutathione Synthetase (GSS)*	Antioxidant defenses
*Nuclear Factor, Erythroid 2 Like 2 (NFE2L2)*	Antioxidant defenses
*Tumor Necrosis Factor (TNF)*	Inflammation
*Interleukin 1 Beta (IL1B)*	Inflammation
*(Interleukin 6 (*/*IL6)*	Inflammation
*C-X-C Motif Chemokine Ligand 8 (CXCL8)*	Inflammation
*Nuclear Factor Kappa B Subunit 1 (NFKB1)*	Inflammation
*Hypoxia Inducible Factor 1 Subunit Alpha (HIF1A)*	Inflammation
*Peroxisome Proliferator Activated Receptor Gamma (PPARG)*	Inflammation
*BCL2 Associated X, Apoptosis Regulator (BAX)*	Autophagy, senescence and cell death
*Tumor protein p53 (TP53)*	Autophagy, senescence and cell death
*B cell leukemia*/*lymphoma 2 (BCL2)*	Autophagy, senescence and cell death
*Beclin 1 (BECN1)*	Autophagy, senescence and cell death
*Cyclin-dependent kinase inhibitor 1 (CDKN1A)*	Autophagy, senescence and cell death
*Cyclin Dependent Kinase Inhibitor 2A (CDKN2A)*	Autophagy, senescence and cell death
*Lysosomal Associated Membrane Protein 2 (LAMP2)*	Autophagy, senescence and cell death
*Sequestosome 1 (SQSTM1)*	Autophagy, senescence and cell death
*Parkin RBR E3 Ubiquitin Protein Ligase (PARK2)*	Autophagy, senescence and cell death
*Microtubule-associated proteins 1A*/*1B light chain 3B (MAP1LC3B)*	Autophagy, senescence and cell death
*Mechanistic Target Of Rapamycin Kinase (MTOR)*	Autophagy, senescence and cell death
*Lamin B1 (LMNB1)*	Autophagy, senescence and cell death
*Pumilio RNA Binding Family Member 1 (PUM1)*	

**Table 2 molecules-27-06183-t002:** List of primers used in this work.

Gene	Design RefSeq	Fwd Primer	Rev Primer
CDKN1A	NM_001291549.N	TGGAGACTCTCAGGGTCGAAAA	CGGCGTTTGGAGTGGTAGAA
CDKN2A	NM_001195132.N	CACCGCTTCTGCCTTTTCA	CCCACATGAATGTGCGCTTA
COL1A1	NM_000088.3	CCCAAAGGATCTCCTGGTGAA	GCCAGGGCTTCCAGTCA
COL3A1	NM_000090.N	CTCCTGGAAAGAATGGTGAAAC	GTCCTGTGTCTCCTTTGTCA
CS	NM_004077.N	GGCCCAATGTAGATGCTCAC	CCCAAACAGGACCGTGTAGTA
ELN	NM_001278939.N	CTGCTAAGGCAGCTGCAAA	CGTAAGTAGGAATGCCTCCAAC
TBP	FLDM-001376.1	CGAATATAATCCCAAGCGGTTTGC	AGCTGGAAAACCCAACTTCTGT
BCL2	FLDM-006133.1	CCCGCGACTCCTGATTCATT	AGTCTACTTCCTCTGTGATGTTGT
PUM1	FLDM-007129.1	GCAAAGATGGACCAAAAGGA	ATTGGCTGGGAAACTGAATG
TP53	FLDM-011174.1	GGAGCACTAAGCGAGCACTG	GGAACATCTCGAAGCGCTCA
PARK2	FLDM-011496.1	GTGTTTGTCAGGTTCAACTCCA	GAAAATCACACGCAACTGGTC
GABPA	FLDM-011683.1	GGAACAGAACAGGAAACAATG	CTCATAGTTCATCGTAGGCTTA
TFAM	FLDM-014938.1	GTTTCTCCGAAGCATGTG	GGTAAATACACAAAACTGAAGG
SOD1	FLDM-017721.1	CGAGCAGAAGGAAAGTAATG	GGATAGAGGATTAAAGTGAGGA
NRF1	FLDM-018128.1	TTGAGTCTAATCCATCTATCCG	TACTTACGCACCACATTCTC
HPRT1	FLDM-018246.1	CCCTGGCGTCGTGATTAGTG	CGAGCAAGACGTTCAGTCCT
HIF1A	FLDM-018811.1	CAACATGGAAGGTATTGCACTG	ACCAAGCAGGTCATAGGTGG
YWHAZ	FLDM-021001.1	TGTAGGAGCCCGTAGGTCATC	GTGAAGCATTGGGGATCAAGA
PPARGC1A	FLDM-034408.1	GCGAAGAGTATTTGTCAACAG	TTGGTTTGGCTTGTAAGTGT
SIRT3	FLDM-039625.1	CGTCACTCACTACTTTCTCC	GATGCCCGACACTCTCTC
SIRT1	FLDM-042031.1	GTAGGCGGCTTGATGGTAAT	GGGTTCTTCTAAACTTGGACTCT
SQSTM1	FLDM-043352.1	AGAATCAGCTTCTGGTCCATCG	TTCTTTTCCCTCCGTGCTCC
PRKAA1	FLDM-044393.1	TCCGTAGTATTGATGATGAAAT	TTAGGTCAACAGGAGAAGAG
PINK1	FLDM-045703.1	TGTGGAACATCTCGGCAGGT	GGCTAGTTGCTTGGGACCTC
LAMP2	FLDM-048658.1	CTGCCGTTCTCACACTGCTC	ATGCTGAAAACGGAGCCATTAAC
BAX	FLDM-051166.1	AGCTGACATGTTTTCTGACGGCAA	CACAGGGCCTTGAGCACCAG
MAP1LC3A	FLDM-058668.1	CAGCAAAATCCCGGTGAT	CTTGACCAACTCGCTCAT
CYCS	FLDM-062546.1	CGTTGAAAAGGGAGGCAAGC	TCCCCAGATGATGCCTTTGTTC
SOD2	FLDM-062627.1	GAAGTTCAATGGTGGTGGTCAT	TTCCAGCAACTCCCCTTTGG
HMOX1	FLDM-068154.1	CTGCTGACCCATGACACCAA	GGGCAGAATCTTGCACTTTGT
BECN1	FLDM-069595.1	ATCCAGGAACTCACAGCTCCA	TGCCTCCCCAATCAGAGTGA
NQO1	FLDM-071287.1	CTGGAGTCGGACCTCTATGC	GGGTCCTTCAGTTTACCTGTGAT
MTOR	FLDM-080915.1	TCCGAGAGATGAGTCAAGAGG	CACCTTCCACTCCTATGAGGC
NFE2L2	FLDM-090829.1	AACTACTCCCAGGTTGCCCA	AGCAATGAAGACTGGGCTCTC
GLB1	NM_001317040.N	GGTGGGACCAATTTTGCCTA	AGTGGGGCATCATAGTCGTA
GSS	NM_001322495.N	AAAAGGGGTCTCTGGACCAA	GTAGCCATCCCGGAAGTAAAC
IL1B	NM_000576.N	GACCTGAGCACCTTCTTTCC	CGTGCACATAAGCCTCGTTA
IL6	NM_000600.3	AGAGCTGTGCAGATGAGTACAA	GTTGGGTCAGGGGTGGTTA
CXCL8	NM_000584.2	ACACTGCGCCAACACAGAAA	CAGTTTTCCTTGGGGTCCAGAC

## Data Availability

Data supporting the results are available upon reasonable requests.

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
