# Peer review of "Targeting Hydroxybenzoic Acids to Mitochondria as a Strategy to Delay Skin Ageing: An In Vitro Approach"

_molecules, 2022, doi:10.3390/molecules27196183_

Round 1
Reviewer 1 Report
This paper compares the cytotoxicity of four hydroxybenzoic acid-based mitochondria-targeted antioxidants (MitoBENs, MB1-4) and reports that MB2 has the least cytotoxicity and is effective in reducing oxidative stress in human skin fibroblasts. After synthesizing these hybrid compounds, the authors have already published several papers on the biological activity of these compounds, and this study is one of the basic studies to apply the biological activity of these compounds to skin aging prevention. To point out a few important issues:
1. A well-known antioxidant such as MitoQ or MitoTEMP should be compared as a positive control to evaluate the activity of the compounds in this study.
2. The biological activity of the hybrid compounds in this study should be compared with that of the parent compounds, hydoxybenzoic acids.
3. In the absence of in vivo test results, the title of this study is too broad and should be modified to be specific according to the experimental results.
4. Experimental conditions such as initial concentration of compounds and medium should be indicated in the legend of Figure 2.
5, the concentration of hydrogen peroxide should be indicated in Figure 4. All compounds should be additionally compared at 6.3 uM.
6. The horizontal axis title in Figure 5 should be corrected (concentration-> molecule).
7. The concentration of compounds should be individually indicated in Figure 6.
In conclusion, the efficacy and safety of the compound reported in this study are not evaluated as particularly excellent, and the therapeutic window is very narrow, limiting its applicability. In order to claim the superiority of this compound, it is necessary to demonstrate comparative advantage compared to the existing mitochondrial-targeting antioxidants.
Author Response
Reviewer 1:
- A well-known antioxidant such as MitoQ or MitoTEMP should be compared as a positive control to evaluate the activity of the compounds in this study.
- The biological activity of the hybrid compounds in this study should be compared with that of the parent compounds, hydoxybenzoic acids.
We acknowledge the reviewer for these valuable suggestions. In this revision, the authors included the cytotoxic profile of other types of mitochondria-targeted antioxidants: MitoQ and SKQ1 and parent compounds. The Sigma reference of protocatechuic and gallic acids were included on page 17, line 532. The synthesis of MitoQ and SKQ1 was included on page 17, lines 553-554, and the treatment of compounds on Page 18, section 4.3, lines 599-600. The cytotoxic profile of MitoQ, SkQ1 and natural antioxidants are shown on figures S1 and S2 on the supplementary material and described in the results in section 2.3, pages 4 and 5, lines 146-156 and 186-189, respectively.
The following paragraphs were added to the introduction and discussion sections respectively, in order to explain the use of MitoQ and SKQ1 in cosmetics:
- “A comparison was made with parental compounds, and with MitoQ and SkQ1, well-characterised mitochondria-targeted antioxidants which have now commercial applications in cosmetic products [37,38]” on pages 2-3 lines 98-101
- “After, the relationship between MitoBENs structure and their toxicity on NHDF cells was subsequently studied and compared with their respective parental antioxidants and MitoQ and SkQ1, two mitochondria-targeted antioxidants with current cosmetic applications [37,38]” on page 15 lines 434-436.
Pedro Soares was also added as a co-author of this work for synthesising MitoQ and SKQ1.
- In the absence of in vivo test results, the title of this study is too broad and should be modified to be specific according to the experimental results.
We acknowledge the reviewer for this comment. Considering that only in vitro assays were performed, in this revised version, the authors changed the title to the following: “Targeting Hydroxybenzoic Acids to Mitochondria as a Strategy to Delay Skin Ageing: An In Vitro Approach”
- Experimental conditions such as initial concentration of compounds and medium should be indicated in the legend of Figure 2.
We acknowledge the reviewer for noticing this. The authors included the initial and final concentrations of compounds and how these were diluted in the legend of figure 2 on page 4, lines 138-140.
5, the concentration of hydrogen peroxide should be indicated in Figure 4. All compounds should be additionally compared at 6.3 uM.
We acknowledge the reviewer for this comment. In this revision, the authors included the concentration of hydrogen peroxide in the legend of figure 4 on page 6, lines 205-206. In the comet assay, the genotoxic effects of the compounds were compared at the highest concentrations of each one, i.e that showed less than 30 % reduction in cell viability, similarly to the choice of concentrations tested in the AMES assays.
- The horizontal axis title in Figure 5 should be corrected (concentration-> molecule).
The authors performed the modifications on figures 4 and 5.
- The concentration of compounds should be individually indicated in Figure 6.
In this revised version, the authors included the concentration of compounds individually in the legend of figure 6 on page 8, lines 257-258.
In conclusion, the efficacy and safety of the compound reported in this study are not evaluated as particularly excellent, and the therapeutic window is very narrow, limiting its applicability. In order to claim the superiority of this compound, it is necessary to demonstrate comparative advantage compared to the existing mitochondrial-targeting antioxidants.
We acknowledge the reviewer for this comment. In this revised version, the authors compared the cytotoxic profile of MitoBENS with the cytotoxic profile of the parental antioxidants, MitoQ and SkQ1, as described on page 15, lines 434-436. The authors inserted the followed paragraph: “Page 15 lines: 452-459: Comparing the cytotoxic profile of MitoBENs with their respective natural antioxidants, it is notorious that all MitoBENs present a slightly higher toxicity than their natural precursors (Figure S1), as a logical consequence of the inherent toxicity of TPP cations, as previously described in the literature [33]. However, it should be noted that all of them present a much lower toxicity than MitoQ and SkQ1 (Figure S2), which are two of the most known mitochondria-targeted antioxidants [29,32], and present already in commercial cosmetic products.”

Reviewer 2 Report
Dear Authors,
Your manuscript was very adequate written and scientific sounded.
- line 41, describe ROS in full before abbreviation.
- line 60, describe OS in full before abbreviation.
- lines 65-67, rewrite the sentence. It was not clear enough.
- The stability was performed in an aqueous medium aiming to evaluate possible molecule degradation, however, the medium could have been the ones used to process the investigation. Could you justify the water selection? What was the pH? Temperature at condition?
- Curcuma mangga should be in italic.
- present the analytical curves for the quantification of the molecules.
Author Response
Reviewer 2:
- line 41, describe ROS in full before abbreviation.
- line 60, describe OS in full before abbreviation.
- lines 65-67, rewrite the sentence. It was not clear enough.
- Curcuma mangga should be in italic.
We acknowledge the reviewer for this comment. The authors reviewed all of these comments as suggested by the reviewer.
- The stability was performed in an aqueous medium aiming to evaluate possible molecule degradation, however, the medium could have been the ones used to process the investigation. Could you justify the water selection? What was the pH? Temperature at condition?
We thank the reviewer for the observation. The temperature condition was already described in section 4.2.3.2. In the same section, the authors added the pH of the water. The choice of water was simply to mimic the excipient conditions used in formulations for dermal applications.
- present the analytical curves for the quantification of the molecules.
The curves were inserted in supplementary material in Figure S3.

Reviewer 3 Report
Dear authors,
For starters I want to congratulate you for the article.
Overall, the manuscript is complex and contain new valuable and interesting information’s. The abstract is appropriate and the aim of the work clearly established. Regarding the methodology used, I appreciated first of all the very clear structure and the multitude of tests used. The results are presented in a clear manner. Chapter of discussion is drawn up in a clear and logical manner, making reference to the relatively recent literature data.
The chosen study is interesting and complex, however, I have the following remarks:
- Line 103 - 104: This statement is not relevant for the Introduction chapter. I believe that it should be removed.
- Try to improve the quality of the images. Or maybe you can consider a possible division of these graphics (for example to group in images compounds with similar structures, in order to ease the reading of your article and highlight the most important ones).
Author Response
Reviewer 3:
- Line 103 - 104: This statement is not relevant for the Introduction chapter. I believe that it should be removed. We acknowledge the reviewer for this comment. This statement was removed.
- Try to improve the quality of the images. Or maybe you can consider a possible division of these graphics (for example to group in images compounds with similar structures, in order to ease the reading of your article and highlight the most important ones).
For the sake of clarity and to improve the article's readability, the authors show the cytotoxic effects of the catecholic and pyrogallol compounds separately in figure 3 on page 5 instead of including them on a figure that already contained the cytotoxic profile of all MitoBENs.

Round 2
Reviewer 1 Report
Overall, this reviewer is content with the revision. The manuscript was well improved. However, the title is still too broad like a review paper. Please consider a tentative title below:
Hydroxybenzoic acid-based mitochondria-targeted antioxidants exhibit anti-senescence effects in human dermal fibroblasts and a 3-dimensional skin model
Reviewer 2 Report
Dear Authors,
Thank you for addressing all raised questions.